# Unveiling the Multifaceted Roles of ISG15: From Immunomodulation to Therapeutic Frontiers

**DOI:** 10.3390/vaccines12020153

**Published:** 2024-02-01

**Authors:** Enrique Álvarez, Michela Falqui, Laura Sin, Joseph Patrick McGrail, Beatriz Perdiguero, Rocío Coloma, Laura Marcos-Villar, Céline Tárrega, Mariano Esteban, Carmen Elena Gómez, Susana Guerra

**Affiliations:** 1Department of Molecular and Cellular Biology, Centro Nacional de Biotecnología, Consejo Superior de Investigaciones Científicas (CSIC), 28049 Madrid, Spain; enrique.alvarez@cnb.csic.es (E.Á.); laura.sin@ciberinfec.es (L.S.); perdigue@cnb.csic.es (B.P.); lmarcos@cnb.csic.es (L.M.-V.); mesteban@cnb.csic.es (M.E.); 2Department of Preventive Medicine, Public Health and Microbiology, Universidad Autónoma de Madrid, 28049 Madrid, Spain; michela.falqui@uam.es (M.F.); joseph.mcgrail@estudiante.uam.es (J.P.M.); rocio.coloma@uam.es (R.C.); celina.tarrega@uam.es (C.T.); 3Centro de Investigación Biomédica en Red de Enfermedades Infecciosas (CIBERINFEC), Instituto de Salud Carlos III (ISCIII), 28029 Madrid, Spain; 4Department of Microbiology, Icahn School of Medicine at Mount Sinai, New York, NY 10029, USA; 5Global Health and Emerging Pathogens Institute, Icahn School of Medicine at Mount Sinai, New York, NY 10029, USA

**Keywords:** IFN, ISG15, ISGylation, vaccines, inflammation, immunity, adjuvant, cytokines

## Abstract

The Interferon Stimulated Gene 15 (ISG15), a unique Ubiquitin-like (Ubl) modifier exclusive to vertebrates, plays a crucial role in the immune system. Primarily induced by interferon (IFN) type I, ISG15 functions through diverse mechanisms: (i) covalent protein modification (ISGylation); (ii) non-covalent intracellular action; and (iii) exerting extracellular cytokine activity. These various roles highlight its versatility in influencing numerous cellular pathways, encompassing DNA damage response, autophagy, antiviral response, and cancer-related processes, among others. The well-established antiviral effects of ISGylation contrast with its intriguing dual role in cancer, exhibiting both suppressive and promoting effects depending on the tumour type. The multifaceted functions of ISG15 extend beyond intracellular processes to extracellular cytokine signalling, influencing immune response, chemotaxis, and anti-tumour effects. Moreover, ISG15 emerges as a promising adjuvant in vaccine development, enhancing immune responses against viral antigens and demonstrating efficacy in cancer models. As a therapeutic target in cancer treatment, ISG15 exhibits a double-edged nature, promoting or suppressing oncogenesis depending on the tumour context. This review aims to contribute to future studies exploring the role of ISG15 in immune modulation and cancer therapy, potentially paving the way for the development of novel therapeutic interventions, vaccine development, and precision medicine.

## 1. ISG15: General Features and Signalling 

### 1.1. ISG15: General Features

The Interferon Stimulated Gene 15 (ISG15), a key component of the immune system, was one of the first interferon (IFN)-stimulated genes (ISGs) discovered in 1984 studying IFN-treated cells [1], and the first Ubl modifier protein identified, with ~30% amino acid sequence identity to ubiquitin (Ub) [2]. Structurally, ISG15 comprises two Ubl domains arranged in tandem that are linked by a hinge region. This structure includes four β-sheets and a single α-helix per domain, resembling a head-to-tail dimer of Ub with different surface features [3] (Figure 1). 

ISG15 expression is strongly induced by type I IFN and other stimuli, such as type II and III IFN [4], genotoxic stressors [5], pathogen infection [6], lipopolysaccharide [7], and retinoic acid [8] (Figure 2). ISG15 is exclusive to vertebrates, and its primary sequence shows limited conservation across different species. Significant variation is evident when comparing human ISG15 with mammalian and fish species, showing 30–35% identity. Even between two mammal species the identity can be less than 60% [4]. This observation suggests that only certain elements in the ISG15 structure are essential for maintaining its function. ISG15 may display a specialized role in higher eukaryotes, or even several specialized functions in some species, although redundancy cannot be discarded. Recent findings indicate that ISG15 deficiency in human individuals is associated with increased levels of IFN in their blood, resulting in elevated ISG levels. This observation explains the increased resistance to viral infections. Despite this, these individuals are more susceptible to environmental mycobacteria, basal ganglia calcifications, and autoinflammatory diseases [9]. These differences are primarily due to the anti-inflammatory function of human ISG15, which stabilizes the Ub-specific peptidase 18 (USP18) protein, a critical negative regulator of the IFN-I receptor. In contrast, murine ISG15 lacks this interaction, leading to an alternative regulation of IFN. Consequently, ISG15-deficient mice exhibit a divergent phenotype, rendering them more susceptible to infection by various types of viruses [10].

The IFN-induced ISG15 protein controls a plethora of cellular pathways, such as DNA damage response [11], regulation of DNA replication synthesis and stress [12,13], intracellular trafficking [14], modulation of cytoskeleton dynamics [15], autophagy [16], host antiviral response [17], and cancer occurrence and progression [18]. Additionally, ISG15 protein is involved in various mechanisms, such as modulation of immune response [19], vascular remodelling [20], mitochondrial functionality [21], energy metabolism [22,23], and macrophage activation [24]. ISG15 performs these functions through three different modes of action: (i) it can covalently modify proteins in a process known as ISGylation; (ii) it can act intracellularly by a non-covalent mechanism; and (iii) it can be secreted and acts as an extracellular cytokine (Figure 2).

### 1.2. ISGylation 

ISGylation is a reversible post-translational modification (PTM) in which ISG15 is covalently conjugated to newly synthesized target proteins through an enzymatic cascade reaction [25]. This process plays a central role in host cell antiviral immunity [26]. Immature 17 KDa ISG15 is processed by protease cleavage to the mature 15 KDa ISG15 form with the exposure of the C-terminal LRLRGG motif [27], which binds to lysine (K) residues of the target protein resulting in its ISGylation [25]. Previous studies have demonstrated that the mutation of the two glycine (G) residues of the C-terminal motif to alanine (A) results in the formation of a non-conjugable form of ISG15 [28], highlighting its identity with Ub. ISG15 activation, conjugation, and ligation are mediated by different enzyme forms involved in ubiquitylation processes [29]: (i) ISG15-activating enzyme E1 (a Ub-activating enzyme E1-like protein; Ube1L in mice or UBA7 in humans); (ii) ISG15-conjugating enzyme E2 (similar to Ub-carrier protein H8; UbcH8 in mice or UBE2L6 in humans); and (iii) ISG15-ligating enzyme E3 (similar to Homologous to the E6-AP Carboxyl Terminus (HECT) and RCC1 Like domain (RLD) containing E3 Ub protein ligases 5 and 6; HERC6 in mice or HERC5 in humans) [30,31]. To date, three E3 ligases have been involved in ISG15 conjugation to substrates: HERC5, Tripartite Motif Containing 25 (TRIM25) and Ariadne RBR E3 Ubiquitin Protein Ligase 1 (ARIH1). While HERC5 exclusively serves as an E3 ligase for ISG15 mediating the modification of newly synthesized proteins, both TRIM25 and ARIH1, can also operate as E3 ligases for Ub. There are specific proteases known as deISGylating enzymes that catalyse the removal of ISG15 from its conjugated targets allowing the reversible nature of ISGylation. USP18 is the main human protease that cleaves ISG15 [32]; however, the deubiquitinase USP16 has recently emerged as a novel ISG15 cross-reactive protease that may regulate metabolic pathways [33]. This intricate regulatory mechanism enables cells to modulate the potency and duration of ISGylation in response to various cellular and environmental stimuli.

Surprisingly, the functional implications of ISGylation remain largely unknown and subjected to controversy. Some reports suggest that ISG15 may disrupt the function of ISGylated proteins [34], while others have demonstrated that ISGylation has the capability to negatively regulate the turnover of ubiquitylated proteins by the proteasome [35].

Antiviral effect of ISGylation

The canonical role of ISG15 has been widely investigated in the context of bacterial and viral infections. An antiviral effect mediated by ISGylation has been reported in many viral diseases through covalent binding to viral and host target proteins using in vitro and/or in vivo systems [26] (Figure 3, Table 1). As an example, ISGylation of non-structural protein 1 (NS1) from influenza A virus (IAV) results in impaired packaging of viral ribonucleoproteins (RNPs) inhibiting its replication and thus contributing to the antiviral action of IFN [36]. Moreover, the replication and viral spread of respiratory syncytial virus (RSV) [37], human papilloma virus 16 (HPV16) [38], Sindbis virus (SINV) [28], and Ebola virus (EBOV) [39] is altered by ISGylation. Due to the importance of the antiviral response exerted by ISG15, it is not surprising that different viruses, such as influenza B virus (IBV) [40] and severe acute respiratory syndrome coronavirus-2 (SARS-CoV-2) [41] have developed mechanisms to counteract its antiviral effects [26]. Intriguingly, more recent investigations have uncovered a proviral role of ISG15 in the replication of hepatitis B [42] and hepatitis C [43] viruses. The findings indicate that viruses may have evolved alongside their hosts, allowing them to use ISGylation for their own benefit. Considering the emerging knowledge on the antiviral functions of lipid droplets, recent findings highlight that the Ring finger protein 213 (RNF213), implicated in moyamoya arteriopathy (MA), exhibits antimicrobial activity against RSV, Listeria monocytogenes, herpes simplex virus 1 (HSV-1), and coxsackievirus B3 (CVB3). RNF213 is a substrate for ISGylation, forming oligomers on lipid droplets and acting as a sensor for ISGylated proteins [44]. Furthermore, it has been reported that ISGylation of the Stimulator of Interferon Genes (STING) protein facilitates its oligomerization, which is essential for the STING-mediated type I IFN induction in DNA sensing of viruses. In the context of human immunodeficiency virus type 1 (HIV-1) infection, a recent study shows that ISG15 deficiency exacerbates infection rates due to lack of ISGylation of the STING protein and suppression of the STING-dependent DNA sensing pathway [45].

The dual role of ISGylation in cancer

The effects of ISGylation in cancer appear to be multifaceted, exhibiting both suppressive and promoting effects depending on tumour type [48] (Table 2). ISG15 and the enzymes responsible for catalysing ISGylation and deISGylation exhibit dysregulation in different cancer types, such as breast, ovarian, pancreatic, colorectal, glioma, and prostate cancers [49]. ISGylation can induce cancer progression by the modulation of several signalling pathways, such as epidermal growth factor receptor (EGFR) recycling, Ki-Ras, and Akt in breast cancer, suggesting a potential therapeutic avenue by inhibiting ISGylation to reverse the malignant condition of these cells [50]. Additionally, Yes-associated protein (YAP), an effector protein associated with tumour formation, has been shown to play a role in metastasis when it is ISGylated [51]. In the context of pancreatic cancer, ISGylation modulates the activity of key proteins involved in pancreatic cancer stem cells (CSCs) metabolism and survival, suggesting that targeting ISGylation could be a potential therapeutic strategy for pancreatic cancer by disrupting the metabolic plasticity of CSCs and promoting their apoptosis [52]. Interestingly, in hepatocellular carcinoma (HCC) the nuclear factor erythroid 2-like 3 (NFE2L3), which has been largely involved in cancer development, induces the expression of ISG15 by binding to the antioxidant response element (ARE) located in the ISG15 promoter. This, in turn, enhances the ISGylation of p53 in a manner dependent on its transcription factor function, consequently amplifying the proteasome-dependent degradation of ISGylated p53 and promoting tumour cell growth [53].

Moreover, ISGylation plays an antagonistic role in Ub-mediated proteasomal degradation in breast, colorectal, and lung adenocarcinoma tumour cells, where it competes for polyubiquitination binding sites [49]. It has recently been reported that ISG15 selectively targets glycosylated PD-L1, leading to its degradation. This enhances PD-L1-targeted immunotherapy and effectively inhibits tumour growth *in vivo* [54]. Other studies suggest that ISGylation inhibits tumorigenesis by destabilizing growth-regulating proteins [55]. Notably, targeted repression of USP18/UBP43, a regulator associated with ISG15, has been demonstrated to reduce proliferation and increase apoptosis in lung cancer and acute promyelocytic leukemia cell lines [56]. This underscores the potential significance of ISGylation, despite some ambiguity in understanding its precise functional consequences.

Overall, ISGylation emerges as a complex and multifaceted PTM intricately involved in the regulation of cellular processes, antiviral responses, and cancer-related pathways. The reversible conjugation of ISG15 to target proteins, orchestrated by distinct enzymes that resemble those of the ubiquitylation process, underscores its pivotal role in modulating immune responses and cellular functions. The functional implications of ISGylation remain a subject of ongoing research, reflecting the need for a comprehensive understanding of its diverse effects. 

**Table 2 vaccines-12-00153-t002:** ISG15 effects in cancer. The different tumour types, ISG15 target proteins, ISG15 form (conjugated/free intracellular), and its effect (protumour/antitumour) are indicated.

Tumour Type	ISG15 Target	ISG15 Form	Effect	Ref.
Breast cancer	N.S.	Free intracellular	Antitumour	[57]
Breast cancer	N.S.	Conjugated	Protumour	[58]
HCC	p53	Conjugated	Protumour	[53]
LUAD	YAP	Conjugated	Protumour	[51]
Lung adenocarcinoma	Glycosylated PD-L1	Conjugated	Antitumour	[54]
Pancreatic cancer	ERK1/2 phosphorylation	Free extracellular	Protumour	[52]
PDAC	N.S.	Free intracellular/ extracellular	Protumour	[59]
Solid tumor	HIF1α	Conjugated/ Free intracellular	Antitumour	[60]

Abbreviations: N.S.: Not specified in the cited bibliography; HCC: Hepatocellular carcinoma; p53: Tumour protein p53; LUAD: Lung adenocarcinoma; YAP: Yes-associated protein; PD-L1: Programmed death-ligand 1; ERK1/2: Extracellular signal-regulated kinase 1/2; PDAC: Pancreatic ductal adenocarcinoma; HIF1α: Hypoxia inducible factor 1 subunit alpha.

### 1.3. Intracellular Free-ISG15 Functions

The intracellular functions of free-ISG15 have been shown to broadly impact on cell functionality, exhibiting both stimulatory and inhibitory effects [4]. Notably, free-ISG15 binds to the USP18 protein in human cells, stabilizing and preventing its degradation [61]. This interaction reinforces the USP18-mediated inhibition of the IFN receptor signalling pathway. In contrast, the murine form of ISG15 does not stabilize murine USP18 in mouse cells, although it maintains the IFN inhibition activity [10]. Beyond USP18/free-ISG15-mediated stabilization, another mechanism of IFN-I regulation is executed by free-ISG15 through its interaction with Leucine (L)-rich repeat-containing protein 25 (LRRC25)-retinoic acid-inducible gene-I (RIG-I) protein-p62. This interaction mediates autophagic degradation of RIG-I, which is critical for IFN downregulation mediated by LRRC25 [62]. Additionally, free-ISG15 plays a role in mediating hypoxic and inflammatory responses. Monomeric ISG15 interacts with Hypoxia inducible factor 1α (HIF1α), preventing its dimerization with HIF1β and the subsequent downregulation of HIF-1α-mediated gene expression and tumorigenic growth [60]. As we discussed above, free-ISG15 has also demonstrated to exert antiviral activity (Figure 3 and Table 1) in EBOV-like particle virus budding [45], inhibiting the interaction between E3 Ub ligase and E2 conjugation enzyme, and thus preventing ubiquitination, which is crucial for EBOV-like particle virus budding. Moreover, monomeric ISG15 regulates the autophagic clearance of proteins, particularly those that are ISGylated [39], by binding to Histone deacetylase 6 (HDAC6) and Ub-binding protein p62 [63]. Recent in vitro studies further indicate that free-ISG15 plays a role in inhibiting pseudorabies virus (PRV), acting as a positive regulator of IFN-I by facilitating the activation of Interferon regulatory factor 3 (IRF3) and promoting the release of IFN-I. Additionally, ISG15 accelerates the activation and nuclear translation of Signal transducers and activators of transcription 1 (STAT1) and 2 (STAT2), facilitating the formation of the STAT1/STAT2/IRF9/Interferon stimulated gene factor 3 (ISGF3) complex and the induction of IFN-stimulated response elements (ISRE) genes [47]. In vitro evidence demonstrated ISG15 dimer formation when high monomeric ISG15 levels were detected within the cells. C78 residue, localized in the hinge region of human ISG15 (C76 and C143 in mouse), is responsible for dimer formation through disulphide bonds [64], resulting in a decreased availability of monomeric ISG15 for ISGylation. When these residues are nitrosylated, a PTM that is activated by the same stimuli that activate ISG15 transcription [65] results in an increased availability of monomeric ISG15 disposal for ISGylation. These data indicate that nitrosylation may contribute to the ISGylation antiviral pathway [64]. 

The intracellular monomeric ISG15 plays a dual regulatory role in tumour cells (Table 2). Unconjugated ISG15 increases major histocompatibility complex (MHC) expression on the surface of breast cancer cells exhibiting anti-tumour properties [57] while elevated ISG15 levels are associated with adverse factors in breast cancer, such as higher histological grade, lymphovascular invasion, larger tumour size, hormone receptor negativity, and Human epidermal growth factor receptor 2 (HER2) positivity [66]. Additionally, free-ISG15 promotes cancer stem cell-like features in pancreatic ductal adenocarcinoma (PDAC), with this effect being regulated by TRIM29 and CAPN3 [59].

The multifaceted roles of intracellular free-ISG15 open avenues for the development of therapeutic interventions. Understanding its impact on cellular functions, particularly in the context of cancer and antiviral responses, provides the optimal background for the design of targeted therapies to modulate immune responses, inhibit tumour progression, and counteract viral infections.

### 1.4. ISG15 as an Extracellular Cytokine

Unconjugated ISG15 actively contributes to the innate immune response through its extracellular signalling function (Figure 4), a phenomenon initially identified in 1991 [67]. ISG15 is secreted by various cell types, including neutrophils, fibroblasts, lymphocytes, and monocytes. The presence of an Ub-like structure in ISG15 raises curiosity regarding its extracellular signalling activity, and the lack of a signal peptide adds complexity to the understanding of its secretion mechanism [3]. Efforts in past years demonstrated its release in a non-canonical manner, including exosomes [68], through neutrophil granules or via apoptosis [69].

Due to the use of the Ub-Activated Interaction Traps (UBAITs), Lymphocyte function-associated antigen 1 (LFA–1) was identified as a surface receptor of ISG15. LFA-1 is expressed on B cells, T cells, neutrophils, macrophages, and Natural killer (NK) cells [70]. ISG15 interacts with the αL domain (CD11a) of LFA-1, the binding site of Intercellular Adhesion Molecule 1 (ICAM1), the best characterized ligand of the LFA-1 receptor [71], essential for the recruitment and adhesion of immune cells to antigen-presenting cells (APC), and for cell killing induced by NK cells. The interaction domain of ICAM1 requires magnesium ion coordination residues of CD11a domain of LFA-1 for its binding [72]. These residues did not affect ISG15 binding, demonstrating that these two molecules did not compete in vitro for the αL binding. ISG15 can act directly on LFA-1, triggering Steroid Receptor Coactivator (SRC) kinase family activation, which is implicated in integrin signalling [73].

Extracellular ISG15 stimulates type II interferon (IFN–γ) and IL-10 production synergically with IL-12 in lymphocyte cell lines. A proposed rationale for the synergistic effect with IL-12 is that IL-12 triggers the transcriptional activation of *IFN-γ* gene, while ISG15 operates post-transcriptionally, likely influencing cytokine-containing secretory vesicles to enhance SRC family kinases (SFK)-dependent IFN-γ and IL-10 secretion [74]. Additionally, recent evidence highlights ISG15-mediated induction of several other cytokines, such as CXCL1, CXCL5, tumour necrosis factor (TNF), IL-1, and IL-6 [75]. Moreover, extracellular ISG15 has been reported to function as a chemotactic factor for neutrophils [76], inducing NK cell proliferation [77] and e-cadherin formation on dendritic cells (DC) [78]. 

Extracellular free-ISG15 has the potential to enhance the immune system exerting an anti-tumour effect not only by activating the innate and adaptive arms of the immune response but also by upregulating p53 expression, thus promoting apoptosis of tumour cells [79] (Table 2). Research by Burks et al. revealed that extracellular ISG15 hinders breast cancer cell growth and boosts NK cell migration to tumours [57] whereas Huggins et al. demonstrated heightened expression of ISG15-related genes in certain macrophages within the murine breast cancer microenvironment [80]. However, it cannot be asserted that free-ISG15 always plays an anti-tumour role as it has also been observed to have a pro-oncogenic effect by increasing ERK1/2 phosphorylation in pancreatic cancer, promoting tumour stemness [52]. Predictably, combination treatments targeting both tumour microenvironment and the tumour itself hold promise for clinical efficacy. 

## 2. ISG15 as an Adjuvant in Vaccines

This section will explore the exciting prospects and emerging research regarding the impact of the adjuvant properties of ISG15 in enhancing immune responses. ISG15 capacity to amplify and modulate immune responses will be described, highlighting its promising role in vaccine development and therapeutic interventions.

The antiviral immune response is essential for controlling infection and preventing viral replication in the host. In this sense, cellular immunity mediated by cytotoxic T cells (CTLs) is essential for the elimination of virus-infected cells. However, CTL response can be suboptimal in some cases, leading to virus persistence and inadequate antiviral immunity. Recent studies have identified the ISG15 protein as a potentially effective immune adjuvant against different antigens [81,82]. In a pioneering study, Villarreal et al. evaluated the role of the wild-type (ISG15GG) and the mutated (ISG15AA) forms of ISG15 as adjuvants for a DNA vaccine targeting the E6 and E7 proteins of HPV16. The ISG15AA protein is unable to ISGylate target proteins. They observed a significant increase in the E7-specific IFN-γ responses in groups receiving either ISG15GG or ISG15AA compared to the administration of the antigen alone. Both experimental groups showed higher frequencies of effector CD8^+^ T cells secreting proinflammatory cytokines and expressing degranulation markers. No significant differences were detected between the groups immunized with ISG15GG or ISG15AA, suggesting that the adjuvant effect of ISG15 could be independent of its ISGylation capacity [81]. In C57BL/6 mice subcutaneously transplanted with TC-1 tumour cells, the combined administration of HPV vaccine candidate with ISG15GG or ISG15AA significantly reduced the tumour growth and increased the lifespan up to 42 days after tumour implantation, ISG15AA being the form that exhibits the most promising results, with 6 out of 10 mice tumour-free at day 42. The anti-tumour effect of ISG15 was directly associated with the cytotoxic potential of antigen-specific stimulated CD8 T cells. The potential of ISG15 as an alarm signalling molecule to activate NK cells and promote CD8 T cell responses was further reinforced by Iglesias-Guimarais et al. The combined administration of a DNA vaccine candidate against HPV with a plasmid containing the *ISG15* gene by intraepidermal route induced greater cell migration and recruitment, as well as increased proinflammatory activity in the inoculation area. Both forms of ISG15 (ISG15GG and ISG15AA) enhanced the specific CTL response against HPV viral antigen, promoting the induction of cytotoxic T lymphocytes, both short-lived effector cells (SLECs) and memory precursor effector cells (MPECs). As the previous study, these findings confirm that the potential role of ISG15 as an immune adjuvant is independent of its ability to ISGylate proteins. In addition, they demonstrated that the clonal expansion of CD8^+^ T cells induced by ISG15 was independent of CD4^+^ T cells. However, co-administration of the vaccine candidate with a plasmid containing the ISG15 mutant Y94L-Q100D, which is unable to bind to its receptor LFA-1, completely abolished its ability to improve the immune response, suggesting that this immune enhancement is mediated by ISG15 through LFA-1 signalling [82].

The adjuvant role of ISG15 protein has not only been evaluated in response to oncogenic viruses. Recently, the potential of ISG15GG or ISG15AA was assessed in combination with HIV-1 vaccine candidates [83]. Intramuscular administration of a DNA vector expressing ISG15AA induced greater infiltration of immune cells into the muscle tissue compared to ISG15GG. However, differences in cellular recruitment in the proximal draining lymph nodes between both ISG15 forms were not observed. Co-administration of ISG15 with a DNA plasmid expressing the HIV-1 envelope protein gp120 in the prime followed by a Modified Vaccinia virus Ankara (MVA) vector expressing the HIV-1 Env and Gag-Pol-Nef antigens in the boost (MVA-B), greatly increased the magnitude and polyfunctional profile of effector CD8^+^ T cells, with ISG15GG showing a better immunostimulatory profile (Figure 5A). This contrasts with previous studies, suggesting that the ISGylation process that occurs in the cell may play an important role in the development of an effective antigen-specific cellular response. Additionally, co-administration of ISG15 allowed for a reduction in the dose of DNA-gp120 by up to 5 times without producing significant differences in the development of HIV-1 Env-specific immune responses [83]. This aspect could be the result of an enhancement in the DNA sensing due to the co-administration of ISG15, which is able to increase the oligomerization of STING. The process is crucial for the effective induction of type I IFN during the sensing of viral DNA. By facilitating STING oligomerization, ISG15 could potentially improve the cell ability to recognize and respond more efficiently to viral DNA, thereby strengthening the antiviral immune response.

Differences detected in previous studies regarding where the immunological potential of ISG15 lies, whether in its ability to ISGylate proteins or in its role as an extracellular signalling molecule, still need to be elucidated. While the studies by Villarreal et al. and Iglesias-Guimarais et al. [81,82] support the idea that the adjuvant potential of ISG15 is independent of its ISGylation activity, the study by Gómez et al. [83] does not rule out the possibility that the canonical PTM function of ISG15 could be relevant in the induction of adaptive-specific cellular responses. Current studies are trying to clarify this controversial aspect using different approaches and expression platforms. A recent study by Falqui et al. aimed to investigate the effect of expressing wild-type or mutated forms of ISG15 from an MVA vector in combination with the HIV-1 vaccine candidate MVA-B on the immune response in a mouse model (Figure 5B). Results demonstrated that the MVA vector expressing the mutated ISG15 (ISG15AA) compared to the wild-type ISG15 (ISG15GG) induced a higher IFN-I production, indicating a more robust innate immune response, and increased the magnitude and polyfunctional profile of the HIV-1-specific CD8 T cells [84]. The adjuvant activity of ISG15 was not only related to its extracellular immunoregulatory activity since we have observed that the endogenous ISGylation was also enhanced after the overexpression of the mutant ISG15AA by the MVA vector. Overall, although differences in immunization regimen, administration route, or vector platform could be responsible for the differential behaviour of ISG15 forms, it seems that both cytokine and ISGylation activities can synergistically contribute to the enhancement of the cellular specific immune responses. Therefore, further studies should be conducted to elucidate the specific factors that are influencing the diverse responses observed, providing a comprehensive understanding of the effect of the different ISG15 forms in various immunization contexts. These investigations will contribute to refine the application of ISG15 as an adjuvant and to optimize its potential therapeutic benefits in diverse vaccination strategies.

The exploration of the potential adjuvant effect of ISG15 and its therapeutic implications requires further investigation. Critical questions remain regarding the optimal model systems for distinguishing the functions of free versus conjugated ISG15. There are significant differences in viral immunity between mice and humans, raising the question of how these differences influence the effects of ISG15. As we discussed above, studies using ISG15–/– mice have demonstrated increased susceptibility and replication rates for IAV, IBV, HSV-1, SINV, and murine gammaherpesvirus 68 (MHV68) compared to wild-type mice [85] but not for vesicular stomatitis virus (VSV) and lymphocytic choriomeningitis virus (LCMV) [86]. Neonatal mice lacking ISG15, when challenged with chikungunya virus (CHIKV), exhibited increased pathogenesis compared to wild-type mice [87]. Interestingly, this enhanced pathogenesis was not attributed to increased viral replication but rather to a cytokine storm phenomenon, similar to the increased mortality observed in ISG15–/– mice infected with a mutant vaccinia virus (VACV) lacking the E3 protein, that normally binds to ISG15 and undermines its restrictive ability [88]. Clinical studies, however, present a different scenario. Human patients with inherited ISG15 deficiency develop type I interferonopathy [61], yet serological tests reveal an enhanced and prolonged resistance to a diverse range of DNA and RNA viruses, accompanied by a reduction in severity, viral replication, and susceptibility to infection [61]. Nevertheless, as we mentioned previously, it is essential to note that a small cohort of human patients exhibiting genetic susceptibility to mycobacterial diseases, with homozygous genetic lesions abrogating ISG15 expression, displayed enhanced susceptibility to Salmonella and Mycobacterium infections due to IFN-gamma deficiency [9]. Thus, the divergent effects of ISG15 in mice and humans underscore the potential influence of biochemical and functional distinctions between human and murine ISG15.

## 3. ISG15 as a Therapeutic Target in Cancer Treatment

This section will try to decipher the intricate connections between ISG15 and cancer pathways, exploring the potential use of ISG15 in innovative cancer therapies. ISG15 clearly displays double-edged roles in malignant cancers, and a careful evaluation of risk-benefit balance should be performed prior to the administration of available ISG15-targeted cancer therapy. In various studies, increased expression of ISG15 has been observed in certain types of cancers [89,90,91,92], and this elevation has been associated with tumour progression. The specific consequences of ISG15 overexpression in these cancers can vary and depend on the tumour microenvironment, as well as on the interactions with other molecules and cellular pathways. However, in many cases, the increase of ISG15 is associated with properties that promote proliferation, invasion, and resistance to programmed cell death, contributing to tumour progression [49,50,93]. Intriguingly, it does not only promote tumour growth, invasion, and metastasis, making it an attractive target for intervention, but also functions as an immunomodulatory antitumour factor [94]. Consequently, ISG15 should be considered as a double-edged sword in cancer treatment. This section will try to decipher the intricate connections between ISG15 and cancer pathways, exploring the potential use of ISG15 in innovative cancer therapies. From underlying molecular mechanisms to promising preclinical studies, the multifaceted role of ISG15 in cancer therapy is highlighted, offering new perspectives of targeted interventions against malignancies.

The potential of ISG15 as a therapeutic target for cancer using active vaccination is being explored in different models. Studying breast cancer, Wood et al. evaluated the antitumoral effect of a vaccine candidate based on Listeria monocytogenes vector that expresses ISG15 (Lm-LLO-ISG15) in BALB/c and FVB/NJ mice previously implanted with 4T1/Luc and NT2 mammary tumours, respectively, and in a transgenic FVB/N HER2/neu mouse model which develops autochthonous mammary tumours without prior therapeutic intervention at 4 months of age. Mice vaccinated intraperitoneally with three doses of Lm-LLO-ISG15 showed a significant reduction in tumour size, as well as an inhibition of the tumour growth in the syngeneic 4T1/Luc and NT2 mammary tumour models. In FVB/N HER2/neu mice the development of autochthonous tumour was significantly delayed after Lm-LLO-ISG15 vaccination, with more than 80% of the mice being healthy after 49 weeks from birth [95]. The therapeutic potential of Lm-LLO-ISG15 vaccine has also been assessed in murine tumour models of renal cell carcinoma and colorectal carcinoma, two types of cancer where ISG15 is overexpressed as a tumour-associated antigen correlating with disease progression and metastasis [96,97]. In the syngeneic renal cell carcinoma, the intraperitoneal administration of three doses of the vaccine candidate significantly reduced tumour growth and increased survival similarly to current treatments based on the administration of sunitinib and anti-PD-1 therapy [96]. The intraperitoneal administration of Lm-LLO-ISG15 also significantly increased the infiltration of effector T cells in the tumour and reduced tumour growth in subcutaneous and orthotopic murine models of colorectal carcinoma after three doses of the vaccine candidate [97]. In all these studies (Figure 6), the protection conferred by the Lm-LLO-ISG15 vaccine crucially depends on CD8^+^ T cells and the immune cell recruitment into the tumour microenvironment (TME). Depletion of CD8 T cells abrogates the anti-tumour efficacy achieved by vaccination, indicating that the expansion of vaccine-induced ISG15-specific CD8 T cells can directly inhibit tumour growth overexpressing ISG15 as a tumour-associated antigen [95,96,97]. 

Administration of exogenous ISG15 has also been evaluated in cancer therapy. A single subcutaneous dose of purified free-ISG15 was able to suppress tumour growth and increase NK cell infiltration in xenografted breast tumours in nude mice [57]. Another study conducted in a mouse xenograft model of adenocarcinoma revealed that after subcutaneous administration of three doses of purified free-ISG15 per week for a period of 5 weeks the tumour size was significantly reduced and the area of tumour necrosis was quite larger, indicating inhibition of tumour growth and metastasis [98]. 

Despite these encouraging findings, it is crucial to clarify the dual role of ISG15 in oncogenesis, exhibiting both promoting and suppressing effects in different tumour systems. 

## 4. Conclusions and Future Perspectives

Here we have elucidated the diverse roles of ISG15 in cellular processes, ranging from immune response modulation to its emerging significance in cancer therapy. The intricate interplay between ISGylation and different cellular pathways underscores its dynamic nature and multifunctional attributes. As we navigate the complex landscape of ISG15, it becomes evident that its manipulation could be a useful strategy, particularly in the context of vaccine development and cancer treatment.

The exploration of ISG15 adjuvant properties has unveiled promising prospects for enhancing HPV- and HIV-1-specific immune responses. The reviewed studies indicate that both wild-type (ISG15GG) and mutated (ISG15AA) forms of ISG15 exhibit notable adjuvant effects, leading to heightened cytotoxic T cell responses and improved vaccine efficacy. However, a nuanced understanding of ISG15 mechanisms remains elusive, with debates surrounding the significance of its ISGylation capacity. While certain investigations propose independence from ISGylation for the adjuvant effect, others suggest its potential relevance in inducing specific cellular responses. This unresolved issue underscores the need for further research to dissect the precise factors influencing the diverse immune responses observed. Future studies should focus on elucidating the intricate interplay between cytokine and ISGylation activities of ISG15. Clarifying the context-specific roles of ISG15 in diverse immunization scenarios, considering factors such as immunization schedule, administration routes, or vector platforms, will be crucial. These investigations will not only refine the application of ISG15 as an adjuvant but also optimize its therapeutic benefits in different vaccination strategies.

Further research is also needed to decipher the precise mechanisms underlying the involvement of ISG15 in immune signalling and tumorigenesis. Exploring the potential crosstalk between ISG15 and other cellular components may unveil novel therapeutic targets and intervention strategies. Future investigations should be aimed to fill current knowledge gaps, paving the way for innovative therapeutic approaches. By deciphering the complexities of ISG15-mediated processes, researchers will be able to develop novel strategies for drug development and precision medicine, strengthening the advances in both immunology and oncology fields.

## Figures and Tables

**Figure 1 vaccines-12-00153-f001:**
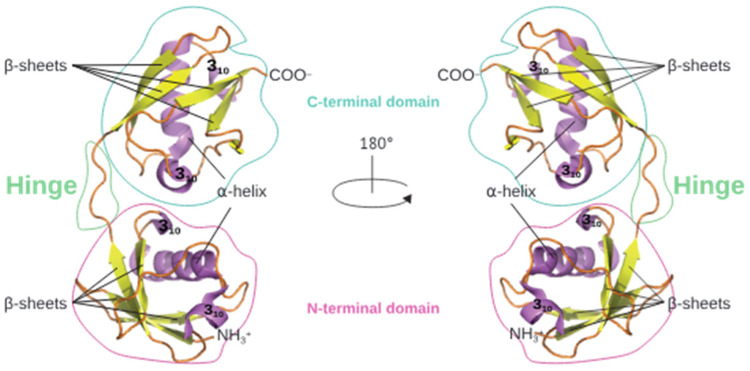
Human ISG15 structure. Left: Ribbon diagram of ISG15 with two Ubl domains linked by a hinge region (PDB entry 1ZM2). The two Ubl domains are conformed by four β-sheets and one α-helix. Right: The same structure rotated 180°. The C-terminal domain is indicated in blue, and the N-terminal domain is depicted in pink. Adapted from Ref. [3].

**Figure 2 vaccines-12-00153-f002:**
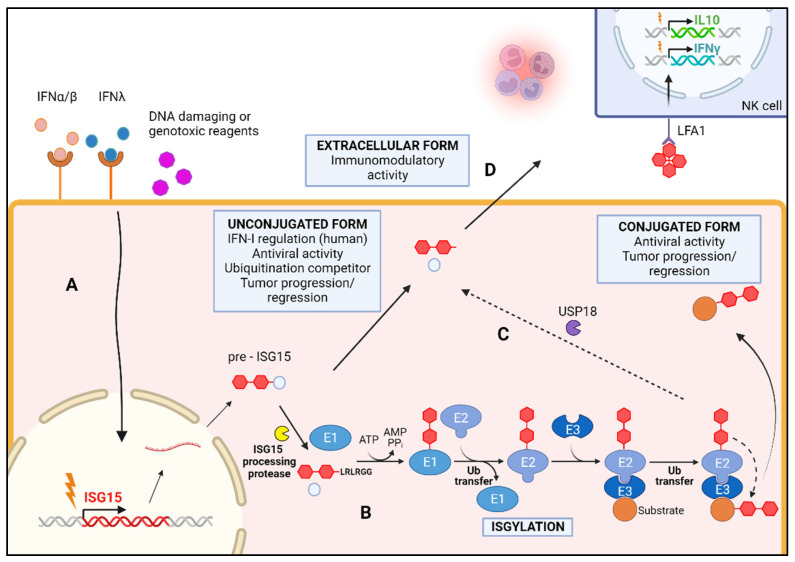
Induction and functions of ISG15. (**A**) Different stimuli, such as the presence of type I and III IFNs or genomic damage, induce the expression of ISG15. (**B**) The immature precursor of ISG15 is processed into its mature form, capable of conjugating with other proteins through a process called ISGylation. (**C**) ISG15 can be removed from its target by the action of deubiquitinases such as USP18. (**D**) Unconjugated ISG15 can be released into the extracellular environment acting as immunomodulator. Created with BioRender.com.

**Figure 3 vaccines-12-00153-f003:**
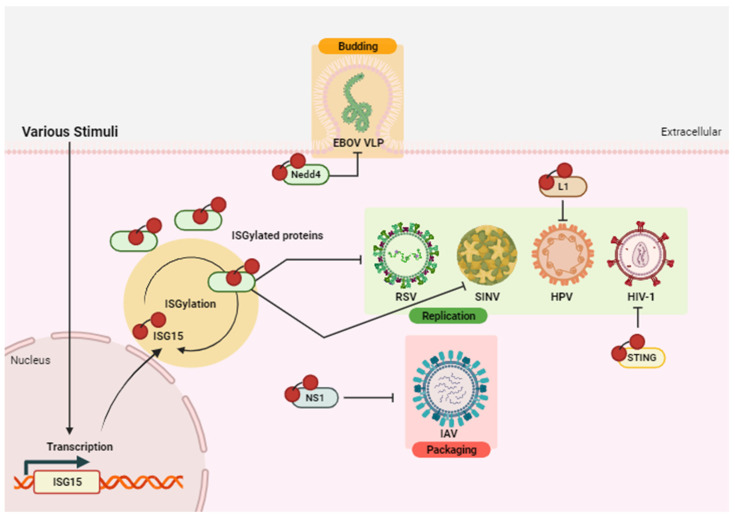
ISGylation antiviral activity. Different stimuli, such as microbial infections, IFNs and genomic damage, induce the expression of ISG15. This molecule conjugates with several target proteins (viral and host origin) through a reversible process called ISGylation. An antiviral effect mediated by ISGylation has been reported. ISGylation inhibits RSV and SINV replication in a yet-to-be-clarified manner, and HPV and HIV-1 replication through the respective viral Major capsid protein L1 (L1) and the cellular STING proteins. Additionally, ISGylation can block IAV packaging acting on the viral protein NS1 and viral budding of EBOV VLP modifying Nedd4. Created with BioRender.com.

**Figure 4 vaccines-12-00153-f004:**
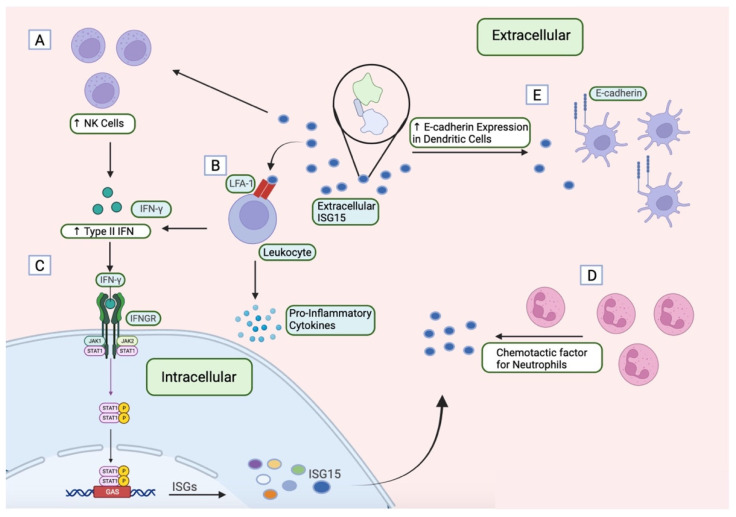
Schematic representation of ISG15 extracellular signalling functions. (**A**) Extracellular ISG15 induces NK cell proliferation. (**B**) Extracellular ISG15 binds to LFA-1 receptor in leukocytes triggering the production of type II IFN and pro-inflammatory cytokines. (**C**) Type II IFN activates JAK/STAT pathway, inducing STAT1 phosphorylation, which binds to GAS, and production of ISGs. (**D**) Extracellular ISG15 acts as a chemotactic factor for neutrophils. (**E**) ISG15 in its extracellular form leads to elevated e-cadherin expression in dendritic cells. GAS: Gamma Activated Sequence; IFN: Interferon; IFNGR: Interferon Gamma Receptor; ISG: Interferon Stimulated Gene; JAK: Janus Kinase; LFA-1: Lymphocyte Function-Associated Antigen 1; STAT: Signal Transducers and Activators of Transcription; NK: Natural killer. Created with BioRender.com.

**Figure 5 vaccines-12-00153-f005:**
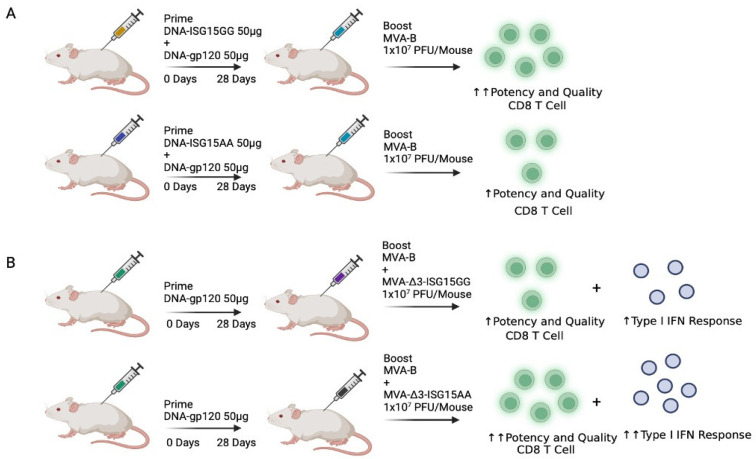
Immunization regimens in mice using ISG15 as an adjuvant in combination with HIV-1 vaccine candidates. (**A**) Mice were primed with 50 µg of DNA-ISG15GG (able to perform ISGylation) or DNA-ISG15AA (unable to perform ISGylation) + 50 µg of DNA-gp120, followed by a boost with 1 × 10^7^ PFU/mouse of MVA-B at 28 days post-prime. The administration of DNA-ISG15GG in the prime increased the potency and quality of the HIV-1-specific CD8 T cells compared to DNA-ISG15AA [83]. (**B**) Mice were primed with 50 µg of DNA-gp120, followed by a boost with 1 × 10^7^ PFU/mouse of MVA-B + MVA-Δ3-ISG15GG or MVA-Δ3-ISG15AA at 28 days post-prime. The immunization with MVA-Δ3-ISG15AA induced higher magnitude and quality of the HIV-1-specific CD8 T cells along with an increased type I IFN response compared to MVA-Δ3-ISG15GG [84]. IFN: Interferon; MVA: Modified Vaccinia virus Ankara; PFU: Plaque Forming Unit. Created with BioRender.com.

**Figure 6 vaccines-12-00153-f006:**
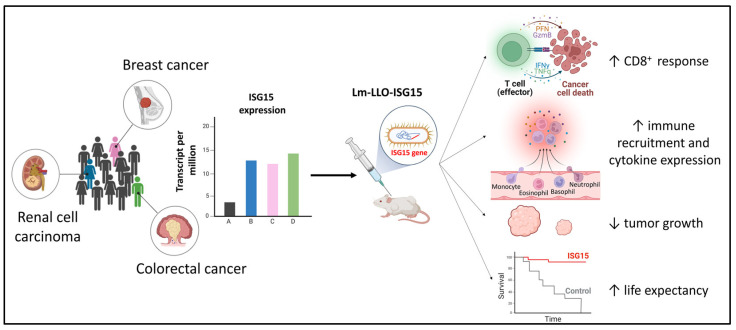
ISG15 in tumorigenesis and cancer therapy. The expression of ISG15 is increased in breast, renal, and colorectal cancers [90,91,92]). Immunogenicity studies using the Lm-LLO-ISG15 vaccine candidate in the mouse model against these types of cancer and its effect in the generated immune response are indicated. Created with BioRender.com.

**Table 1 vaccines-12-00153-t001:** ISG15 antiviral/proviral activity. The different viruses, ISG15 target proteins (from host/viral origin), ISG15 forms (conjugated/free intracellular), its effect (antiviral/proviral), and study type (*in vitro*/*ex vivo*/*in vivo*) are indicated.

Virus	ISG15 Target	ISG15 Form	Effect	Study Type	Refs.
EBOV	Nedd4 (Host)	Conjugated/ Free intracellular	Antiviral	*In vitro*	[39,45]
HBV	HBx (Viral)	Conjugated	Proviral	*In vitro*	[42]
HCV	NS5A (Viral)	Conjugated	Proviral	*In vitro*	[43]
HIV-1	STING (Host)	Conjugated	Antiviral	*In vitro*	[46]
HPV16	L1 (Viral)	Conjugated	Antiviral	*In vitro*	[38]
IAV	NS1 (Viral)	Conjugated	Antiviral	*In vitro*	[36]
PRV	STAT2 (Host)	Free intracellular	Antiviral	*In vitro*	[47]
RSV	N.S.	Conjugated	Antiviral	*In vitro*, *ex vivo*	[37]
SINV	N.S.	Conjugated	Antiviral	*In vivo*	[28]

Abbreviations: EBOV: Ebola virus; Nedd4: Neuronal precursor cell-expressed developmentally downregulated 4; HBV: Hepatitis B virus; HBx: Hepatitis B protein x; HCV: Hepatitis C virus; NS5A: Non-structural protein 5A; HIV-1: Human immunodeficiency virus type 1; STING: Stimulator of interferon genes; HPV16: Human papilloma virus 16; L1: Major capsid protein L1; IAV: Influenza virus A; NS1: Non-structural protein 1; PRV: Pseudorabies virus; STAT2: Signal transducer and activator of transcription 2; RSV: Respiratory syncytial virus; N.S.: Not specified in the cited bibliography; SINV: Sindbis virus.

## Data Availability

Not applicable.

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
