# Peer review of "Unveiling the Multifaceted Roles of ISG15: From Immunomodulation to Therapeutic Frontiers"

_vaccines, 2024, doi:10.3390/vaccines12020153_

Round 1

Reviewer 1 Report

Comments and Suggestions for Authors

The manuscript is arranged logically and is written in a well-organized and comprehensive way even for those who are not so familiar with the particulate topic. Especially with these readers in mind, I suggest to significantly expand the data on phenotype of patients born with inactive mutation of ISG15 (as briefly mentioned in lines 58 – 62). the symptom manifestation is quite wide, and this information would be appropriate. As well, description of ISG 15-deficient mice should be included. A more detailed discussion on the different role of ISgG15 in anti-viral immune responses (and possibly other mechanisms) in mice and humans is desirable, despite the fact, that literature covering the topic is rather limited.

The authors should consider moving the text in paragraph beginning at line 423 to the introductory part of the Chapter 3.

Please increase the fonts in figures, as some of the labels are hard to read (e.g., Figure 4).

Author Response

Reviewer 1

The manuscript is arranged logically and is written in a well-organized and comprehensive way even for those who are not so familiar with the particulate topic. Especially with these readers in mind, I suggest to significantly expand the data on phenotype of patients born with inactive mutation of ISG15 (as briefly mentioned in lines 58 – 62). The symptom manifestation is quite wide, and this information would be appropriate. As well, description of ISG15-deficient mice should be included. A more detailed discussion on the different roles of ISG15 in anti-viral immune responses (and possibly other mechanisms) in mice and humans is desirable, despite the fact that literature covering the topic is rather limited. 

The authors should consider moving the text in paragraph beginning at line 423 to the introductory part of the Chapter 3. 

Please increase the fonts in figures, as some of the labels are hard to read (e.g., Figure 4).

We highly appreciate the reviewer for the positive overall point of view of the manuscript.

In detail:

Comment: I suggest to significantly expand the data on phenotype of patients born with inactive mutation of ISG15 (as briefly mentioned in lines 58 – 62). The symptom manifestation is quite wide, and this information would be appropriate. As well, description of ISG15-deficient mice should be included.

Response: Following reviewer´s comments, we have now expanded the description of the phenotype of patients with inactive mutations of ISG15. We have included detailed information about the broad manifestation of symptoms in these individuals, specifically addressing the increased resistance to viral infections, as well as the increased susceptibility to environmental mycobacteria, basal ganglia calcifications, and autoinflammatory diseases. These details are presented with relevant references, emphasizing the anti-inflammatory function of ISG15 in stabilizing Ub-specific peptidase 18 (USP18) in humans.

We have also included additional information on ISG15-deficient mice, explaining that murine ISG15, in contrast to human ISG15, does not interact with the USP18 protein. This leads to alternative regulation of IFN and a divergent phenotype in ISG15-deficient mice, making them more susceptible to various viral infections.

Lines 60-67:This observation explains the increased resistance to viral infections. Despite this, these individuals are more susceptible to environmental mycobacteria, basal ganglia calcifications, and autoinflammatory diseases [1]. These differences are primarily due to the anti-inflammatory function of human ISG15, which stabilizes the Ub-specific peptidase 18 (USP18) protein, a critical negative regulator of the IFN-I receptor. In contrast, murine ISG15 lacks this interaction, leading to an alternative regulation of IFN. Consequently, ISG15-deficient mice exhibit a divergent phenotype, rendering them more susceptible to infection by various types of viruses [2]”.

Comment: A more detailed discussion on the different roles of ISG15 in anti-viral immune responses (and possibly other mechanisms) in mice and humans is desirable, despite the fact that literature covering the topic is rather limited.

Response: As suggested by the reviewer, the expanded discussion now includes a detailed comparison of viral immunity in mice and humans, highlighting significant differences. Specifically, we have presented results from studies in ISG15-/- mice demonstrating increased susceptibility and replication rates for several viruses. In addition, we commented the cytokine storm phenomenon observed in neonatal mice lacking ISG15 and its implications. The revised manuscript now delves into clinical studies, contrasting the effects of ISG15 deficiency in human patients with those observed in mice. We have highlighted the divergent effects of ISG15 in mice and humans, emphasizing biochemical and functional distinctions between human and murine ISG15.

Lines 404-427: “The exploration of the potential adjuvant effect of ISG15 and its therapeutic implications requires further investigation. Critical questions remain regarding the optimal model systems for distinguishing the functions of free versus conjugated ISG15. There are significant differences in viral immunity between mice and humans, raising the question of how these differences influence the effects of ISG15. As we discussed above, studies using ISG15-/- mice have demonstrated increased susceptibility and replication rates for IAV, IBV, HSV-1, SINV and murine gammaherpesvirus 68 (MHV68) compared to wild-type mice.[3] but not for vesicular stomatitis virus (VSV) and lymphocytic choriomeningitis virus (LCMV) [4]. Neonatal mice lacking ISG15, when challenged with chikungunya virus (CHIKV), exhibited increased pathogenesis compared to wild-type mice [5]. Interestingly, this enhanced pathogenesis was not attributed to increased viral replication, but rather to a cytokine storm phenomenon, similar to the increased mortality observed in ISG15-/- mice infected with a mutant vaccinia virus (VACV) lacking the E3 protein that normally binds to ISG15 and undermines its restrictive ability. [6]. Clinical studies, however, present a different scenario. Human patients with inherited ISG15 deficiency develop type I interferonopathy [7], yet serological tests reveal an enhanced and prolonged resistance to a diverse range of DNA and RNA viruses, accompanied by a reduction in severity, viral replication, and susceptibility to infection [7]. Nevertheless, as we mentioned previously, it is essential to note that a small cohort of human patients exhibiting genetic susceptibility to mycobacterial diseases, with homozygous genetic lesions abrogating ISG15 expression, displayed enhanced susceptibility to Salmonella and Mycobacterium infections due to IFN-gamma deficiency [1]. Thus, the divergent effects of ISG15 in mice and humans underscore the potential influence of biochemical and functional distinctions between human and murine ISG15.”

Comment: The authors should consider moving the text in paragraph beginning at line 423 to the introductory part of the Chapter 3. 

Response: Following reviewer´s recommendation, we have now moved the indicated paragraph to the introductory part of the Chapter 3. 

Comment: Please increase the fonts in figures, as some of the labels are hard to read (e.g., Figure 4).

Response: As suggested by the reviewer, we have now increased the font size of the figures, particularly focusing on Figure 4 (now Figure 5) with the aim to enhance the overall readability and clarity of the figures for the readers.

Reviewer 2 Report

Comments and Suggestions for Authors

The review entitled Unveiling the Multifaceted Roles of ISG15: From Immunomodulation to Therapeutic Frontiers by Álvarez et al. provides a comprehensive summary of the functions of the ISG15 protein and ISGylation in immune regulation, antiviral defense, and tumor development. Below are some suggestions for improvement on this manuscript for the authors' consideration:

1. There is an error in the description in lines 59-60. Generally, individuals with inactivated ISG15 gene mutations should be sensitive to viral infections rather than developing resistance.

2. The title of Figure 2 is inappropriate. It is suggested to change it to "Induction and Functions of ISG15".

3. In section "1.2 ISGylation", it is recommended to summarize the role of ISGylation in antiviral defense and tumor development in the form of a table. This could include information such as the types of viruses and tumors, targets of ISGylation, and the resulting effects. Additionally, it can also be combined with the functions of intracellular or free ISG15.

4. Regarding the antiviral function of ISG15, it is suggested to create a figure summarizing its role. This figure could describe the antiviral action of ISG15 based on different stages of viral infection and replication.

5. Figure 3 is overly simplistic and does not effectively illustrate how ISG15 modulates the immune system, such as the receptors on different immune cells and the signaling pathways activated.

These suggestions aim to enhance the clarity and depth of the manuscript, providing a more comprehensive understanding of the multifaceted roles of ISG15.

Author Response

Reviewer 2

The review entitled  “Unveiling the Multifaceted Roles of ISG15: From Immunomodulation to Therapeutic Frontiers” by Álvarez et al. provides a comprehensive summary of the functions of the ISG15 protein and ISGylation in immune regulation, antiviral defense, and tumor development. Below are some suggestions for improvement on this manuscript for the authors' consideration:

  1. There is an error in the description in lines 59-60. Generally, individuals with inactivated ISG15 gene mutations should be sensitive to viral infections rather than developing resistance.
  2. The title of Figure 2 is inappropriate. It is suggested to change it to "Induction and Functions of ISG15".
  3. In section "1.2 ISGylation", it is recommended to summarize the role of ISGylation in antiviral defense and tumor development in the form of a table. This could include information such as the types of viruses and tumors, targets of ISGylation, and the resulting effects. Additionally, it can also be combined with the functions of intracellular or free ISG15.
  4. Regarding the antiviral function of ISG15, it is suggested to create a figure summarizing its role. This figure could describe the antiviral action of ISG15 based on different stages of viral infection and replication.
  5. Figure 3 is overly simplistic and does not effectively illustrate how ISG15 modulates the immune system, such as the receptors on different immune cells and the signaling pathways activated.

These suggestions aim to enhance the clarity and depth of the manuscript, providing a more comprehensive understanding of the multifaceted roles of ISG15.

We are thankful to the reviewer for the highly valuable comments and suggestions.

In detail:

Comment 1: There is an error in the description in lines 59-60. Generally, individuals with inactivated ISG15 gene mutations should be sensitive to viral infections rather than developing resistance.

Response: The information presented is consistent with established research. Regarding the antiviral role, Bogunovic´s lab published a paper indicating that ISG15 deficiency increases viral resistance in humans but not in mice (2). We have now explained the underlying mechanisms contributing to this resistance.

Lines 60-67:This observation explains the increased resistance to viral infections observed in these individuals. At the same time, however, these individuals exhibit increased susceptibility to environmental mycobacteria, basal ganglia calcifications, and autoinflammatory diseases [1]. These differences are primarily due to the anti-inflammatory function of human ISG15, which stabilizes the Ub-specific peptidase 18 (USP18) protein. In contrast, murine ISG15 lacks this interaction, leading to an alternative regulation of IFN. As a result, ISG15-deficient mice exhibit a divergent phenotype, rendering them more susceptible to infection by various types of viruses [2]”.

Comment 2: The title of Figure 2 is inappropriate. It is suggested to change it to "Induction and Functions of ISG15".

Response: As suggested by the reviewer, we have now changed the title of Figure 2 to "Induction and functions of ISG15".

Comment 3: In section "1.2 ISGylation", it is recommended to summarize the role of ISGylation in antiviral defense and tumor development in the form of a table. This could include information such as the types of viruses and tumors, targets of ISGylation, and the resulting effects. Additionally, it can also be combined with the functions of intracellular or free ISG15.

Response: We are very grateful to the reviewer for this comment, as the inclusion of this information has greatly improved the manuscript. The first table (Table 1) outlines the dual roles of ISG15 in antiviral defense and proviral activity, including different types of viruses, targeted molecules, resulting effects and type of study. The second table (Table 2) comprehensively describes the effects of ISG15 in tumorigenesis, providing a concise summary of its impact on various tumour types.

Comment 4: Regarding the antiviral function of ISG15, it is suggested to create a figure summarizing its role. This figure could describe the antiviral action of ISG15 based on different stages of viral infection and replication.

Response: We are again very grateful to the reviewer for this suggestion, as the inclusion of this new figure (Figure 3) has greatly improved the manuscript by creating an illustrative figure that succinctly outlines the multiple antiviral roles of ISG15 and provides a comprehensive visual representation of the involvement of ISG15 in combating viral threats.

Comment 5: Figure 3 is overly simplistic and does not effectively illustrate how ISG15 modulates the immune system, such as the receptors on different immune cells and the signaling pathways activated.

Response: We agree with the reviewer and, following his suggestion, we have expanded Figure 3 (now Figure 4). The revised figure now represents a more comprehensive scenario of how ISG15 modulates the immune system, highlighting the receptors on different immune cells and the pathways that are activated. We believe this improved version of the figure provides a more detailed and accurate understanding of the mechanisms involved in ISG15 extracellular signalling.

  1. Hermann, M.; Bogunovic, D. ISG15: In Sickness and in Health. Trends Immunol 2017, 38, 79–93, doi:10.1016/j.it.2016.11.001.
  2. Speer, S.D.; Li, Z.; Buta, S.; Payelle-Brogard, B.; Qian, L.; Vigant, F.; Rubino, E.; Gardner, T.J.; Wedeking, T.; Hermann, M.; et al. ISG15 Deficiency and Increased Viral Resistance in Humans but Not Mice. Nat Commun 2016, 7, doi:10.1038/ncomms11496.

Reviewer 3 Report

Comments and Suggestions for Authors

The manuscript by Enrique Álvarez et al explores the role of ISG15 in immune modulation and cancer therapy, with the potential aim for the development of novel therapeutic interventions and vaccine development. The manuscript is interesting and well presented. As possible integration, it would be interesting to mention the connection between the ISGylation of the Stimulator of Interferon Genes (STING) protein and the property of Sting agonists that promote adjuvant effects, discussing the implication of mechanisms  of the reported adjuvant activity.

Author Response

Reviewer 3

The manuscript by Enrique Álvarez et al. explores the role of ISG15 in immune modulation and cancer therapy, with the potential aim for the development of novel therapeutic interventions and vaccine development. The manuscript is interesting and well presented.

Comment: As possible integration, it would be interesting to mention the connection between the ISGylation of the Stimulator of Interferon Genes (STING) protein and the property of STING agonists that promote adjuvant effects, discussing the implication of mechanisms of the reported adjuvant activity.

Response: We are grateful to the reviewer for this valuable suggestion. As suggested, we have now included information that delves into the context of HIV infection, elucidating how ISG15 deficiency may exacerbate infection rates due to the lack of ISGylation of the STING protein, thereby suppressing the STING-dependent DNA sensing pathway. Furthermore, we have discussed the potential implications of this mechanism in the observed adjuvant activity, emphasizing its relevance to our exploration of immune modulation and cancer therapy.

Lines 139-142: “In the context of human immunodeficiency virus type 1 (HIV-1) infection, a recent study shows that ISG15 deficiency exacerbates infection rates due to lack of ISGylation of the STING protein and suppression of the STING-dependent DNA sensing pathway [8]”.

Lines 360-365: “This aspect could be as result of an enhancement in the DNA sensing due to the co-administration of ISG15 which is able to increase the oligomerization of STING. The process is crucial for the effective induction of type I IFN during the sensing of viral DNA. By facilitating STING oligomerization, ISG15 could potentially improve the cell ability to recognize and respond more efficiently to viral DNA, thereby strengthening the antiviral immune response”.

  1. Lin, C.; Kuffour, E.O.; Fuchs, N. V; Gertzen, C.G.W.; Kaiser, J.; Hirschenberger, M.; Tang, X.; Xu, H.C.; Michel, O.; Tao, R.; et al. Regulation of STING Activity in DNA Sensing by ISG15 Modification. Cell Rep 2023, 42, 113277, doi:10.1016/j.celrep.2023.113277.